# Investigating the Formation of Structural Elements in Proteins Using Local Sequence-Dependent Information and a Heuristic Search Algorithm

**DOI:** 10.3390/molecules24061150

**Published:** 2019-03-22

**Authors:** Alejandro Estaña, Malik Ghallab, Pau Bernadó, Juan Cortés

**Affiliations:** 1LAAS-CNRS, Université de Toulouse, CNRS, 31400 Toulouse, France; aestana@laas.fr (A.E.); malik@laas.fr (M.G.); 2Centre de Biochimie Structurale. INSERM, CNRS, Université de Montpellier, 34090 Montpellier, France; pau.bernado@cbs.cnrs.fr

**Keywords:** proteins, structural elements, conformational transitions, structural database, heuristic search algorithms

## Abstract

Structural elements inserted in proteins are essential to define folding/unfolding mechanisms and partner recognition events governing signaling processes in living organisms. Here, we present an original approach to model the folding mechanism of these structural elements. Our approach is based on the exploitation of local, sequence-dependent structural information encoded in a database of three-residue fragments extracted from a large set of high-resolution experimentally determined protein structures. The computation of conformational transitions leading to the formation of the structural elements is formulated as a discrete path search problem using this database. To solve this problem, we propose a heuristically-guided depth-first search algorithm. The domain-dependent heuristic function aims at minimizing the length of the path in terms of angular distances, while maximizing the local density of the intermediate states, which is related to their probability of existence. We have applied the strategy to two small synthetic polypeptides mimicking two common structural motifs in proteins. The folding mechanisms extracted are very similar to those obtained when using traditional, computationally expensive approaches. These results show that the proposed approach, thanks to its simplicity and computational efficiency, is a promising research direction.

## 1. Introduction and Related Work

Proteins are bio-macro-molecules that perform essential functions in living organisms. They are composed of chains of amino acid residues (in the following, we will use the word *residue* to refer to an *amino acid residue*), also called polypeptide chains, that, in most of the cases, fold into functional three-dimensional structures. The amino acid sequence determines the three-dimensional structure and its stability. The sequence also determines the frequency and the transition rate between unfolded and folded states. Understanding the mechanisms of protein folding and unfolding as a function of the amino acid sequence is of paramount importance, giving their relevance in biological processes [1]. Furthermore, numerous diseases are related to the inability of proteins to fold correctly or to form insoluble amyloidogenic aggregates due to mutations or metabolic deregulation [2,3].

Intensive research efforts over several decades, using both experimental and computational approaches, have yielded important bricks of knowledge on the underlying mechanisms of protein folding, unfolding and other conformational transitions [4,5,6,7,8,9]. Nevertheless, we still lack of a complete understanding of these mechanisms. Some theories about protein folding give more importance to interactions between the protein side-chains, whereas others consider that the propensity of protein backbone fragments to form secondary structural elements, such as α-helices, β-sheets and turns, is the most important mechanism for protein folding. Note that, in addition to their importance in the overall protein folding process, small structural elements may play key roles in molecular recognition in intrinsically disordered proteins (IDPs). These elements, the so called molecular recognition elements (MOREs), are partially folded fragments inserted into otherwise disordered chains [10,11]. MOREs recognize with high specificity their globular partners while displaying a moderate affinity, explaining their fundamental role in signalling, metabolic regulation and homeostasis [12].

We believe that local, sequence-dependent structural preferences are essential to drive the formation of structural elements, while other phenomena such as hydrophobic effects or electrostatic forces help in stabilizing the overall structure. Following this hypothesis, we propose a theoretical approach to compute conformational transitions using local structural information extracted from experimental data. Interactions between distant residues are (explicitly) neglected for the exploration of transition paths, with the exception of collisions that would lead to unrealistic conformations. However, as further explained below, non-bonded interactions associated with local structural preferences are implicitly considered, and can be propagated along the sequence thanks to the application of constrains within the path search algorithm.

Information extracted from experimentally determined protein structures is frequently used in computational biology. The usual usage is the prediction of the conformation of the protein side-chains, using the so-called rotamer libraries [13], which encode the most frequent values of the side-chain dihedral angles for each amino acid type. The construction of protein backbone structural databases is less straightforward than for the side-chains as it requires to subdivide proteins into fragments. The length of the fragments and considerations regarding the amino acid sequence may depend on the specific application. Statistics about the most frequent values of the backbone dihedral angles of amino acid types have been frequently used to explore the conformational sampling of highly-flexible proteins or regions [14,15,16]. However, such minimalistic single-residue fragments neglect the effects exerted by neighboring residues. Structural libraries involving larger fragments (usually, from 3 to 14 residues) have been shown to be powerful tools for the prediction of probable (stable) conformations of globular proteins and peptides [17,18,19,20]. Fragment libraries can also be used to investigate conformational transitions in proteins. In a recent work, local moves using a fragment library were combined with other types of structural perturbations to compute transitions between several folded states of a protein [21]. Since the aforementioned fragment libraries were mainly conceived for protein structure prediction, they are focused on the most probable conformations of small and medium-sized fragments. As a consequence, they are not exhaustive enough for the study of conformational transitions. This limitation is more evident when the length of the fragments increases. Fragments involving three consecutive amino acid residues (called tripeptides from now on) represent a good trade-off between sequence-dependent structural preferences and exhaustiveness. Indeed, tripeptides contain relevant structural information [22] and are sufficiently small to capture the conformational variability of the 20 proteinogenic amino acids in their sequence context. Recently, we showed that an extensive database of tripeptides allows to accurately sample the conformational variability of IDPs [23]. Here, we exploit the combination of this type of local structural information with a path search algorithm to compute conformational transitions in small proteins and protein fragments corresponding to relevant structural elements.

A protein cannot exhaustively explore its huge conformational space to seek transition pathways. This idea, referred to as the Levinthal’s paradox [24,25], is widely accepted. Indeed, a protein performs some search process to find the most efficient folding and transition pathways. We can say that the protein follows a powerful heuristic to avoid exploring an astronomically large number of possible pathways. This heuristic is not well understood yet, but, as mentioned above, we believe that local sequence-dependent structural preferences play an important role in it. Our contribution investigates this open question, and proposes a simple, heuristically-guided search algorithm, inspired from artificial intelligence (AI) and robotics, to compute conformational transitions. AI and robotics planning representations and techniques have been found valuable for solving several computational biology problems [26,27,28]. This paper illustrates through an original approach their effectiveness in modeling folding mechanisms of structural elements in proteins.

The approach presented herein is very different from the ones in related work. First, the structural information is collected and used in a different way, and secondly, the algorithmic approach is totally different. Concretely, we use a heuristically guided depth-first algorithm, adapted from search techniques in constraint satisfaction problems over finite sets (CSP) and in automated task planning [29]. In our case, the state variables are the protein tripeptides, which range over finite sets of conformations extracted from a global database. The equivalent of an action is a constrained local change in a state variable. The algorithm relies on adjacency graphs of the state variables [30], which are computed at preprocessing time and are essential for efficiently testing the feasibility of transitions and for calculating the heuristic, which is based on statistical physics considerations. Our approach tends to favor paths going through high-density states, which are the most probable ones according to experimental observations recorded in the structural database. In other words, if we assume that the probability of the observed states for each tripeptide follows a Boltzmann distribution, we can say that the path search tends to follow the valleys of the free-energy landscape [31]. The search process also gives priority to short paths, which should correspond to faster transitions. The structural preferences for a tripeptide (i.e., at the state variable level) tend to be propagated along the sequence due to constraints imposed on the bond angles in the state transition validation, which reinforces neighbor-dependent structural preferences encoded in the database (see Appendix A for details). Thus, the path search process incorporates in an implicit way non-local interactions along the sequence such as backbone hydrogen bonds in α-helices.

We applied our approach to two synthetic mini-proteins, Chignolin [32] and DS119 [33], which were particularly designed to fold into well-defined structural motifs present in natural proteins. These two molecules have been investigated in recent years using different methods [34,35]. The results reported in this paper are consistent with respect to those described in related literature, and already show the interest of the proposed approach, which is extremely fast when compared with currently-used computational methods based on molecular dynamics (MD) simulations [36]. Indeed, MD simulations of large-amplitude protein motions require ad-hoc computer architectures [8] or massively-distributed computing [37]. The efficiency of our approach allows to widely investigate, with modest computational resources, the effect of mutations on protein folding and unfolding, or on other functionally-important conformational transitions.

## 2. Results and Discussion

This section presents results obtained with the proposed approach for the analysis of the folding process of two synthetic mini-proteins, Chignolin and DS119, which were designed to fold into structural motifs present in natural proteins. First, we present a deeper analysis for Chignolin and two point mutants. Then, results presented for DS119 show that the approach is general and can be applied to the investigation of different structural elements.

### 2.1. Chignolin

Chignolin is a synthetic polypeptide consisting of 10 residues [32]. Despite its small size, Chignolin behaves as a macromolecular protein from structural and thermodynamic points of view: it folds into a well-defined structure in water, and shows a cooperative thermal transition between unfolded and folded states [38]. The folded conformation of Chignolin corresponds to a β-hairpin motif, which can be found in many natural proteins (Figure 1d). Therefore, elucidating the folding mechanism of Chignolin helps to understand the folding patterns of more complex proteins. This has motivated several experimental and computational studies on Chignolin in recent years. Here, we compare our results with those of Enemark et al. [34], which are based on extensive molecular dynamics simulations, and provide detailed information at the single-residue level.

Table 1 provides the number of conformations (i.e., number of values of state variables) contained in our database for the eight overlapping tripeptides composing Chignolin. The search space size is upper-bounded by $\prod_{i}\left|D_{i}\right|\approx 4\times 10^{23}$, which is huge when compared to the extremely focused explorations performed by our algorithm. Thanks to the search guidance of its heuristics, we observed a manageable complexity growth, as explained in Section 3.3 and in the Appendix A.

In a first experiment, we assessed the ability to obtain realistic conformations of Chignolin using the structural information encoded in our tripeptide database. We generated an ensemble of 1000 Chignolin states by randomly sampling values of the state variables one by one, in an incremental manner, enforcing the consistency with neighbor state variables, and rejecting those leading to collisions between atoms. Interestingly, several states in this relatively small ensemble are close to the folded conformation of Chignolin [32]. Indeed, 240 over the 1000 sampled states have an angular root-mean-square deviation (RMSD) distance to the folded conformation below 0.5 radian, the closest one being around 0.2 radians (see Figure 1d). This confirms that the most important regions of the conformational space can be sampled by building states from the tripeptide database.

In order to better characterize the conformational ensemble, secondary structure types for each state were identified at the single residue level using define secondary structure of proteins (DSSP) [40]. DSSP distinguishes eight types of structural classes, labeled with a letter: H for α-helix, B for β-bridge, E for strand, G for helix-3, I for helix-5, T for turn, S for bend, and “blank” (here labeled as L) for coil/loop. We used the WebLogo tool [41] to display the structural propensities in the ensemble. WebLogo is usually applied to analyze results of multiple sequence alignment, but it can be used in a different context, as we did. Each logo consists of stacks of symbols, one stack for each position in the sequence. The overall height of the stack indicates the conservation of the DSSP structural class at that position, while the height of symbols within the stack indicates the relative frequency of each class at that position. The results in Figure 1a clearly show the propensity of the central residues to adopt a turn conformation. The rest of the molecule tends to be more extended, although turns are also formed in the C-terminal region. As discussed in detail below, these turns in residues eight and nine play a key role in the folding mechanism of Chignolin. Conversely, turns are not observed in the N-terminal side. These observations are fully consistent with the original study [34], and show that the states sampled using the tripeptide database are structurally relevant.

We repeated the experiment for two mutants of Chignolin: Chignolin-P4A (Pro4 replaced by Ala) and Chignolin-W9A (Trp9 replaced by Ala). Figure 1b shows that, for Chignolin-P4A, the turn propensity slightly decreases in the central region, and that it increases in the N-terminal side. For Chignolin-W9A, Figure 1c shows that the propensity to form turns in the central region is similar to that of the native Chignolin molecule. However, it decreases in the C-terminal region, which may have consequences for the efficiency of the folding process. Overall, these observations are very similar to the results reported in [34], which use computationally expensive molecular dynamics simulations; they show the strong influence of single modifications in the sequence on the conformational preferences of the molecule, and that our approach captures these perturbations.

It has been suggested that the turn in Chignolin originates in the C-terminal region, and then propagates along the chain until reaching the middle residues [34]. This has been called the “roll-up” mechanism. To investigate this mechanism, we selected (among the set of 1000 conformations) 15 conformations of Chignolin presenting turns in residues eight and nine, and with a relatively extended conformation for the rest of the chain. These conformations were used as initial states to compute folding paths, as illustrated in Figure 2. The goal state was defined as the closest conformation to the experimental structure of Chignolin built from values contained in the tripeptide database. These two conformations are very similar, with an angular RMSD of 0.1 radians. The heuristically-guided depth-first search (HDFS) algorithm was applied 20 times to solve each of these 15 problems (i.e., 300 runs in total). On average, the algorithm required around 10 s to find folding pathways (1st column in Table 2), which is extremely fast (CPU time was measured with an Intel^®^ Core^TM^ i7 processor at 2.8 GHz, using a single core (Intel, Santa Clara, CA, USA)). Intermediate states along each path were selected with a step-size corresponding to 1/10th of its total length. The left side panel in Figure 3 shows the structural propensities at the residue level for these intermediate states. It can be observed that the turns in the C-terminal residues tend to disappear, while theses structural elements appear in the middle residues. This “roll-up” mechanism can also be observed in the right side panel in Figure 3, which represents several intermediate states along one of the folding paths. The first frames (starting from the top) show that the curvature of the molecule, initially involving residues eight and nine, rapidly propagates to residues six and seven. Then, residues four and five also bend successively, and the molecule tends to form a hairpin-like structure. Finally, the two terminal parts adopt a relatively extended conformation.

As explained in related work [38], the folding process of Chignolin may lead to misfolded states, which are characterized by interactions between residue pairs Tyr2-Thr8 and Asp3-Gly7, rather than Tyr2-Trp9 and Asp3-Thr8, as in the correctly folded structure. We generated a representative model of a misfolded state, and we computed conformational transitions from initial conformations with the C-terminal turn (C-ter T) to this state. We also computed transitions from fully-extended conformations to folded and misfolded states. The results are summarized in the top part of Table 2. This table provides average values (over 300 runs) for: the computing time required by the HDFS algorithm to find a path; the number of recursions and backtracks; the number of steps in the solution path; the length of the solution path, computed as the sum of the lengths associated to edges in the adjacency graphs; the density of the solution paths, computed as the average of the density of all the state variables along the path. The most meaningful numbers in this table are those associated with the density, since they reflect the probability of existence of each pathway. Compared to the extended→folded pathway, the C-ter T→folded pathway goes across more dense and probable regions. This may explain why Chignolin efficiently folds from unfolded states involving this structural feature. In both cases, starting from C-ter T or fully-extended states, the transitions to misfolded states seem to be much less probable. This may explain why the misfolded state of Chignolin is much less frequently observed than the correctly folded state [42].

We repeated the experiments for the mutant Chignolin-W9A. The results are summarized in the bottom part of Table 2. As mentioned above, the set of conformations generated for these two molecules look structurally similar (see Figure 1 and the associated comments). The figures in Table 2 also show a very similar behavior of the HDFS algorithm when computing transition paths for this mutant compared to the original Chignolin. Interestingly, the main difference is observed for the density of the path extended→misfolded. This path is significantly more favorable in the case of the mutant. Our results complement the study of Enemark et al. [34], which suggested that the replacement of Trp9 by Ala facilitates a “roll-back” mechanism, acting against the “roll-up” mechanism, hindering the formation of the native turn in the middle residues. We show another possible effect of this mutation, favoring the formation of misfolded states in competition with the native structure.

### 2.2. DS119

DS119 is another synthetic polypeptide, consisting of 36 amino acid residues, which was designed to fold into a βαβ motif [33] (see last frame in Figure 4). The folding process of DS119 has been studied using molecular dynamics simulations [35]. This previous work showed that the N-terminal side of the central helix tends to form very quickly. Then, the C-terminal side of the helix starts to form, and the full helix is finally stabilized. The relatively extended fragments at the two ends of the molecule tend to come together at the end of the folding process.

To investigate the folding mechanism of DS119, we applied a similar procedure as for Chignolin. In this case, we selected 15 relatively extended conformations, involving only the L DSSP structural class for all the residues, from a set of 1000 randomly generated conformations using the tripeptide database. These conformations were used as initial states for the HDFS algorithm. As final state, we used the closest conformation to the experimentally solved structure of DS119 (Protein Data Bank (PDB) ID: 2KI0) built from values contained in the tripeptide database. These two conformations are very similar, with an angular RMSD of 0.06 rad. The algorithm was applied 20 times to solve each of these 15 problems (i.e., 300 runs in total).

Figure 4 illustrates the results obtained by the HDFS algorithm. The left side panel shows the evolution of the structural propensities along the folding path, using logos based on DSSP classes. The right side panel represents several intermediate states along one of the solution paths. For clarity purposes, only a few intermediate states are shown using a “cartoon” representation of the backbone, where the helical fragments can be easily identified. It can be observed that, starting from an extended conformation, the protein backbone rapidly starts to bend around residues 12–13. Recall that the S letter, for “bend”, corresponds to a highly curved protein backbone. Hydrogen bonds required to stabilize the helical conformation are not yet identified by DSSP at this early stage. Next, curved/helical fragments started to appear in all central residues (from residue 14 until residue 27), as well as in three residues in the N-terminal side (residues 3–5). The central helix continues to fold, and it is almost completely formed at the 7th intermediate frame. In the final part of the path, the extended fragments at the two ends get close to each other, nearly forming a parallel β-sheet. This description of the folding process strongly resembles the one reported in the literature, based on computationally-expensive simulations [35].

Table 3 presents numbers (averaged over the 300 runs) concerning the performance of the HDFS algorithm to compute folding paths of DS119. The required CPU time (and the number of recursions) is only about three times the one requited to compute folding paths for Chignolin. This shows that, despite the theoretical (worst-case) exponential complexity, in practice, the computing time scales approximately linearly with the number of variables. This tendency has been confirmed by preliminary tests for larger molecules (not presented in this paper). Once again, we insist that computing time is orders of magnitude faster that traditional molecular dynamics simulation methods. The higher density of the path compared to Chignolin can be explained by the lager number of conformations for some of the tripeptides, particularly for those composing the middle helix. Table 4 provides the numbers of conformations (i.e., number of values of state variables) contained in our database for the 34 overlapping tripeptides composing DS119.

## 3. Materials and Methods

The proposed approach relies on a large database of protein structures, represented as sequences of partially overlapping tripeptides. As stressed above, tripeptides are the minimal structurally-relevant units in proteins. The problem is formalized as a search in a space of tripeptide conformations for a feasible path from an initial state to a target state of a protein. The state variables correspond to tripeptides; their values are the conformations of tripeptides actually observed and recorded in the database. A state variable in the sequence describing a protein shares its first two residues with its predecessor and its last two with its successor state variables in the sequence (see details below). A transition between two values of a state variable is feasible if it meets a consistency constraint with respect to the predecessor and successor state variables, and if the corresponding conformation of the protein is collision free. The search algorithm seeks a feasible path using a heuristically-guided depth-first search schema. The heuristic function is a weighted sum of the distance between two conformations, an estimate of the distance to the target and a density term to advantage energetically favorable states.

We present next the construction of the structural database, then the statement of the conformational transition problem as a discrete path search problem; we detail the proposed algorithm and the heuristics used to solve this problem.

### 3.1. Structural Database

A tripeptide database was built from a large set of high-resolution experimentally-determined protein structures. We generated this set from SCOPe (release 2.06) [43], avoiding redundancies in protein sequence and structure. The total number of tripeptides extracted from these protein structures is 5,630,271. The tripeptides are characterized by their amino acid sequence. Since natural proteins involve 20 types of amino acids, the total number of tripeptides is $20^{3}=8000$. The database construction process is illustrated in Figure 5a–c. All the 8000 tripeptides appear in our database. The number of their instances ranges between 9 for the less frequent tripeptide (Cys-Cys-Trp) to 4512 for the most frequent one (Ala-Ala-Ala). The average number of instances is about 688.

It is important to highlight that the database includes fragments extracted from coil regions, which have been shown to be useful elements to model unfolded or disordered proteins [15,23]. Therefore, we assume that the structural information encoded in the database is not limited to folded states, and that it can be useful to investigate folding processes.

We adopt a rigid geometry simplification [44], which assumes constant bond lengths and bond angles. Indeed, the standard deviation for the bond lengths and the bond angles in our database is two orders of magnitude smaller that their average value, and therefore, we can neglect their variation. In addition, as usually done to simplify protein modeling, we assume that the torsion angles corresponding to peptide bonds (i.e., the bonds connecting consecutive residues) are constant. This is also a reasonable assumption given that this angle slightly fluctuates around a value of 0 or π radians (that is, the *cis* and *trans* conformations), with a standard deviation of around 0.1 radians. Therefore the only variables required to determine the conformation of a protein backbone correspond to the ϕ and ψ dihedral angles of each amino acid residue. The database stores these angular values for each tripeptide extracted from the ensemble of protein structures (i.e., six angles for each tripeptide). Figure 6 represents a protein fragment involving five residues, from which three tripeptides are extracted. The angles defining the conformation of each residue are represented on the corresponding bonds.

In this work, we do not consider an all-atom model of the protein side-chains, but a simplified model involving a pseudo-atom for each side-chain. The pseudo-atom is centered at the position of the β-carbon atom, and the size depends on the amino acid type, as originally proposed by Levitt [45]. Therefore, no additional variables are required to represent the side-chains.

Let X be the set of all 8000 tripeptides. An element $x_{i}\in X$ is a state variable in our representation. Let $D_{i}$ be the set of all the conformations of $x_{i}$ recorded in our database. The conformation of $x_{i}$ is characterized by the six backbone dihedral angles of the three residues in the tripeptide, denoted $\phi_{i,j}$ and $\psi_{i,j}$, for 1≤j≤3. Although a conformation is characterized by an angular vector of 6 real numbers, for the purpose of our search algorithm over biologically observed conformations, we consider that the range of each state variable $x_{i}$ is the finite set $D_{i}$ of the recorded conformations in the database. We write $x_{i}=v_{i}$ for some $v_{i}\in D_{i}$.

The distance $d(v_{i},v_{i}^{\prime })$ between two values $v_{i}$ and $v_{i}^{\prime }$ is defined as the angular RMSD between the two corresponding angular vectors. More precisely:$$d\left(v_{i},v_{i}^{\prime }\right)=\sqrt{1/6\sum_{j=1}^{3}\left({\left(\phi_{i,j}-\phi_{i,j}^{\prime }\right)}^{2}+{\left(\psi_{i,j}-\psi_{i,j}^{\prime }\right)}^{2}\right)}$$

We also define the central distance $d_{c}\left(v_{i},v_{i}^{\prime }\right)$ with an identical formula for j=2 solely, i.e., restricted to the central amino acid residue of $x_{i}$. The idea is to compute a feasible path in the conformations of a protein as a sequence of elementary transitions focused on the central residue of each tripeptide.

These distances *d* and $d_{c}$ allows us to structure the finite range $D_{i}$ of each state variable as an adjacency graph, as illustrated in Figure 5d. Its vertices are the elements in $D_{i}$. There is an edge $\left(v_{i},v_{i}^{\prime }\right)$ when $d_{c}\left(v_{i},v_{i}^{\prime }\right)<\theta$ and $d(v_{i},v_{i}^{\prime })<\theta +\xi$, where θ is a variable adjacency threshold and ξ is a small constant tolerance margin. The adjacency threshold θ represents a tradeoff between a fully connected graph (no transition constraints between conformations) and an unconnected one (unreachable conformations), both cases being unrealistic. We set the threshold such that the adjacency graph of each tripeptide has a single connected component with moderate edge connectivity. This threshold θ is slightly different for different tripeptides, with an average value around 1.0 radian. The value of ξ was set to 0.35 radians in all the cases.

The vertices are also characterized by a density function defined as follows:$$\rho \left(v_{i}\right)=1+\left|\left\{v_{i}^{\prime }\;|\;v_{i}^{\prime }\;\mathrm{connected}\;\mathrm{to}\;v_{i}\;\mathrm{and}\;d\left(v_{i},v_{i}^{\prime }\right)<\zeta \right\}\right|.$$

The threshold ζ has to be smaller than the adjacency threshold θ. Here, we set ζ=0.2 radians for all the tripeptides. The density ρ is related to the probability of existence of the corresponding conformation of the tripeptide. Considering basic principles in statistical physics (i.e., the Boltzmann distribution), this probability depends on the energy of the state of the molecule. Thus, the most dense regions in the adjacency graph are also the most energetically-favorable ones.

### 3.2. Formal Statement of the Conformation Path Finding Problem

A protein (or protein region) of interest is defined by a sequence of state variables $\langle x_{1},\ldots ,x_{i},\ldots ,x_{n}\rangle$, with overlaps. For example, the mini-protein Chignolin is a sequence of 10 amino acid residues: 〈Gly-Tyr-Asp-Pro-Glu-Thr-Gly-Thr-Trp-Gly〉; it is defined with eight state variables $x_{1}=$ Gly-Tyr-Asp, $x_{2}=$ Tyr-Asp-Pro, … $x_{8}=$ Thr-Trp-Gly. Hence, the state variables are not independent a transition in a state variable may or may not be consistent with another transition in the previous or following state variables in the sequence.

For a given conformational state of the protein $s=\langle \left(x_{1}=v_{1}\right),\ldots ,\left(x_{i}=v_{i}\right),\ldots ,\left(x_{n}=v_{n}\right)\rangle$, the overlap between consecutive state variables means that a tripeptide $x_{i}$ shares its first two residues with its predecessors in the sequence and its last two with its successors; that is:(1)$$\begin{array}{cccc} \phi_{i,1} & =\phi_{i-1,2}=\phi_{i-2,3},\; & \phi_{i,2}=\phi_{i-1,3}=\phi_{i+1,1},\;\mathrm{and}\; & \phi_{i,3}=\phi_{i+1,2}=\phi_{i+2,1}\;, \end{array}$$ and similarly for the ψ angles.

An elementary state transition with respect to $x_{i}$, from the value $v_{i}$ to an adjacent value $v_{i}^{\prime }$, involves a conformational change mainly in the central residue of $x_{i}$ (by construction of the adjacency graph). This entails constraints on $x_{i-1}$ and $x_{i+1}$ with respect to their current values in state *s*. We express these constraints as inequalities with a tolerance margin as follows:(2)$$\begin{array}{c} |\phi_{i,2}^{\prime }-\phi_{i-1,3}\left|<\epsilon ,\;\;\right|\phi_{i,2}^{\prime }-\phi_{i+1,1}|<\epsilon ,\\ |\psi_{i,2}^{\prime }-\psi_{i-1,3}\left|<\epsilon ,\;\;\right|\psi_{i,2}^{\prime }-\psi_{i+1,1}|<\epsilon . \end{array}$$ where the angles for the last and first residues of $x_{i-1}$ and $x_{i+1}$ correspond to their current values $v_{i-1}$ and $v_{i+1}$. These constraints can be relaxed during the search by dynamically adjusting the value of ϵ, as explained below. Here, we set initially ϵ=0.35 radians.

**Definition** **1** (Feasible transition)**.**
*A transition in the conformation of a protein from a state s where $x_{i}=v_{i}$ to a state $s^{\prime }$ where $x_{i}=v_{i}^{\prime }$ is said to be a feasible transition if and only if:*
*(i)* 
*the values $v_{i-1}$ and $v_{i+1}$ meet the constraints of Equation (Equation 2), and*
*(ii)* 
*there are no collisions between the atoms of the protein in the state $s^{\prime }$.*


*A feasible path is a sequence of feasible transitions.*


Let $\gamma (s,\left(v_{i}\to v_{i}^{\prime }\right))$ denotes the state $s^{\prime }$ corresponding to this transition when it is feasible, otherwise γ is undefined.

The conformation path finding problem can be formally stated as follows: given X and the adjacency graphs of all the state variables in a protein, and given an initial state $s_{0}$ and a goal state $s_{g}$, the problem is to find a feasible path that transforms the protein conformation from $s_{0}$ into $s_{g}$, if there exists such a path.

### 3.3. Search Algorithm

To generate a feasible path from $s_{0}$ to $s_{g}$, we rely on a heuristically-guided depth-first search in the space $\prod_{i}D_{i}$, over all state variables $x_{i}$ in the protein. To ease the presentation, the algorithm is stated in the pseudo-code of Figure 7 as a simple recursive nondeterministic search procedure called HDFS. The initial call is HDFS $\left(s_{0},\langle s_{0}\rangle \right)$. The nondeterministic choice (step labelled ◃) is a convenient notation meaning that the algorithm makes at this point a branching decision; it explores potentially all possible options, expressed here as the set E, and it stops on the first path which succeeds or it returns failure if all paths fail. A metaphor helps explain a nondeterministic specification of an algorithm: it runs as if using a hypothetical machine able to multiply itself at each branching point into identical copies, each copy pursuing the search in parallel until one finds a solution or all fail. The deterministic implementation of HDFS makes at this step a heuristic choice over which it backtracks in case of failure; if needed, this is repeated as long as an option in E remains unexplored. The heuristic driving this choice is detailed below.

The algorithm iterates over all tripeptides in the protein to find their feasible transitions. For a given state variable $x_{i}=v_{i}$ in *s*, the transition-filter procedute checks the values adjacent to $v_{i}$ in graph $D_{i}$. Unfeasible transitions are disregarded, as well as transitions that loop back into a circuit of the search space. The set E is the union of all retained transitions $\left(v_{i}\to v_{i}^{\prime }\right)$ over all state variables. When E is empty, then *s* is a dead end; a backtracking is performed.

In our more efficient and deterministic implementation of the algorithm, E is computed incrementally. E starts with the transitions of a single state variable, which has feasible transitions. E is augmented with respect to new state variables when backtracking requires alternative options. In our current code, the ordering of the state variables in the HDFS loop is not heuristically guided. The effects of state variable ordering heuristics, such as the proximity to the goal or the average density in the adjacency graph, remain to be investigated.

#### 3.3.1. Heuristic Guidance Function

For the results presented in this paper, the search is guided though the ordering in the transition-filter procedure of the set A of feasible values. A is ordered with the following cost function:(3)$$\mathrm{cost}\left(v_{i},v_{i}^{\prime }\right)=d\left(v_{i},v_{i}^{\prime }\right)+w_{1}\times h\left(v_{i}^{\prime },v_{i}^{g}\right)+w_{2}/\rho \left(v_{i}^{\prime }\right),$$ where *d* and ρ are the distance and density functions defined earlier, $v_{i}^{g}$ is the value of $x_{i}$ in the goal state $s_{g}$, *h* is the shortest path in the transition graph to the goal, and $w_{1}$ and $w_{2}$ are weight parameters. The first term seeks to minimize the distance between consecutive states along the path (i.e., to maximize the continuity of the path). The second term is the sum of the distances of a minimal path from $v_{i}^{\prime }$ to the goal. The third term intends to maximize the density of the states along the path, which, as explained earlier, are the most energetically favorable ones. The weights $w_{1}$ and $w_{2}$ permit a tuning of the three components; their proper setting remains to be investigated. Here, we simply set $w_{1}=w_{2}=1$. Note that *h* is a lower bound for the remaining cost from $v^{\prime }$ to $v^{g}$, since a path in the transition graph, minimal with respect to the distance *d*, relaxes the feasibility constraints of Definition 1 and cannot be longer than a feasible path.

In order to speed up the search, a preprocessing of the adjacency graphs labels edges with their distance *d* and computes for every vertex the shortest path to the goal as well as the density of every node in each graph. This is done with a standard graph search algorithm.

The test of collision-free states is computed using a variant of the classical cell linked-list (CLL) algorithm [46]. A pair of non-bonded (pseudo-)atoms is considered to be in collision if their distance is less than 65% of the sum of their radii. In this work, we considered the radii values proposed by Bondi [47] for the backbone atoms, and those proposed by Levitt [45] for the side-chains pseudo-atoms.

Note that the feasibility constraints in Equation (Equation 2) are too conservative. A more flexible definition would also accept as feasible the transitions for which either the current values of $x_{i-1}$ and $x_{i+1}$, or some of their respectively adjacent values $v_{i-1}^{\prime }$ and $v_{i+1}^{\prime }$, meet these constraints. In that case, the state $s^{\prime }=\gamma \left(s,\left(v_{i}\to v_{i}^{\prime }\right)\right)$ involves changes in $x_{i}$ but also in its predecessor and successor state variables. The cost function driving the search would naturally be extended to cost $\left(s,s^{\prime }\right)$ over entire states. Instead, we have implemented a simpler mechanism to locally relax this constraint if the search process gets blocked: if state transitions fail *f* consecutive times (f=5 in our implementation), the tolerance value ϵ is increased to 0.7 radians. ϵ is reset to 0.35 radians after a successful transition. The next section shows that, even with such a simplified implementation, the proposed approach already gives meaningful results.

#### 3.3.2. Properties of HDFS

The algorithm is sound; that is, it returns a path which is feasible, in case of success. This is because each transition meets Definition 1. HDFS is also complete; that is, it finds a feasible path if one exists with respect to the transitions in the adjacency graphs of the state variables. This is the case since in each search state *s*, E is the entire set of feasible transitions over all state variables, loops are avoided, and backtracking is systematic.

As for any backtrack search algorithm, the worst case complexity is exponential, in $O(\prod_{i}|D_{i}|)$. It is possible to compute the total size of the search space for each given problem (using dynamic programming and taking into account state variable dependencies). Nevertheless, this information is not very useful since in practice the algorithm explores a very small fraction of the search space. A more useful complexity model is in $O(d^{b})$, where *d* is the depth of the search (i.e., the length of the found path), and *b* is the branching factor. An upper bound on the branching factor is n×p, where *n* is the length of the protein and *p* is the maximum degree of vertices over all adjacency graphs. However, thanks to the search guidance of its heuristics, we observed a manageable complexity growth. Our experiments with seven proteins, ranging in length 10≤n≤67 residues, show that *b* does not grow with *n*; it is constant and very small, about b≃1.04. The overall search complexity has a low polynomial growth in *n*. Furthermore, we confirmed that, as expected for a local propagation mechanism, the computation time required for each search state is not a function of *n*, but a quite small constant, of about 0.9 ms per state on a standard CPU. The Appendix A details this analysis as well as a discussion contrasting the scalability of our approach with that of MD methods.

## 4. Conclusions

Despite the simplicity of both the algorithm and the heuristic, the results presented in this paper show that the proposed approach constitutes a promising new research direction towards the identification of relevant protein folding pathways. The structural analysis of the folding mechanisms of Chignolin and DS119 are consistent with respect to descriptions provided in the literature. Note however that a more detailed and quantitative comparison between the paths obtained with other methods and trajectories obtained from MD simulations would not be very meaningful, since the aims of both methods are different. The paths provided by our algorithm are an approximation, from which interesting information about folding mechanisms can already be obtained, but that should be refined (using other methods and models) to get access to accurate information at the atomic level (as provided by MD simulations). On the other hand, our algorithm is orders of magnitude faster than atomistic MD simulations.

Overall, the results highlight the importance of local structural preferences, which are encoded in our tripeptide database. They also suggest that interactions between distant residues in the sequence, even though they can be essential for stabilization of the final fold, are less important at an earlier stage to drive the formation of structural elements.

The good results obtained with the implementation presented in this paper motivate us to continue in this research direction. Several points remain to be further investigated. One important question is about the possibility to include non-local interactions in the heuristic cost function. Although this does not seem to be necessary for structural elements or small proteins, interactions between distant residues in the sequence can be essential to study folding processes of larger molecules, or aspects related to stability. We also plan to implement and evaluate transitions over several state variables, as well as different heuristics for variable ordering. More sophisticated, tree-based search algorithms [29] can improve the quality and the diversity of the solutions, particularly for large proteins. Finally, let us mention the limitations imposed by the information contained in the structural database. Structural information is very limited in some regions of the conformational space corresponding to states of low probability, but which may be relevant for an accurate modeling of conformational transitions. With the increasing number of experimentally-determined high-resolution protein structures, we expect that more extensive and higher-quality tripeptide databases will be constructed in the future. Alternatively, these sparsely populated transition regions can be identified using our approach and subsequently explored using physics-based molecular models and (continuous) motion planning algorithms [48].

## Figures and Tables

**Figure 1 molecules-24-01150-f001:**
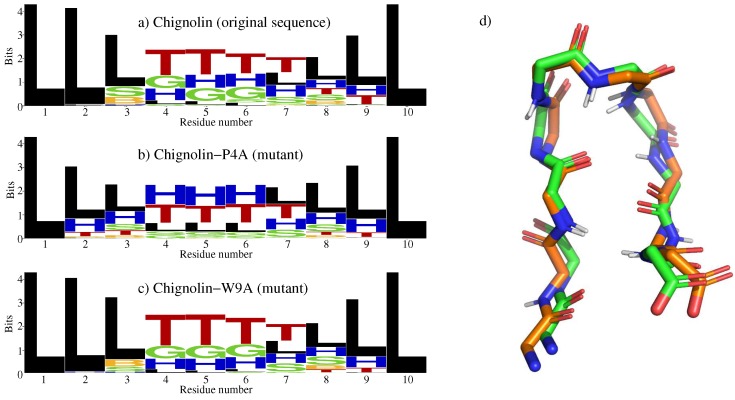
The left side panel represents the structural propensities at the residue level observed from a set of 1000 conformations randomly generated from the structural database. Each plot displays the DSSP structural classes using the WebLogo format for (**a**) Chignolin, and two mutants: (**b**) Chignolin-P4A, and (**c**) Chignolin-W9A. (**d**) Structural representation of Chignolin: superposition of an experimentally determined structure (with carbon atoms in green) and the closest one in the set of 1000 sampled conformations (with carbon atoms in orange). For clarity, only the protein backbone is represented, using PyMOL [39].

**Figure 2 molecules-24-01150-f002:**
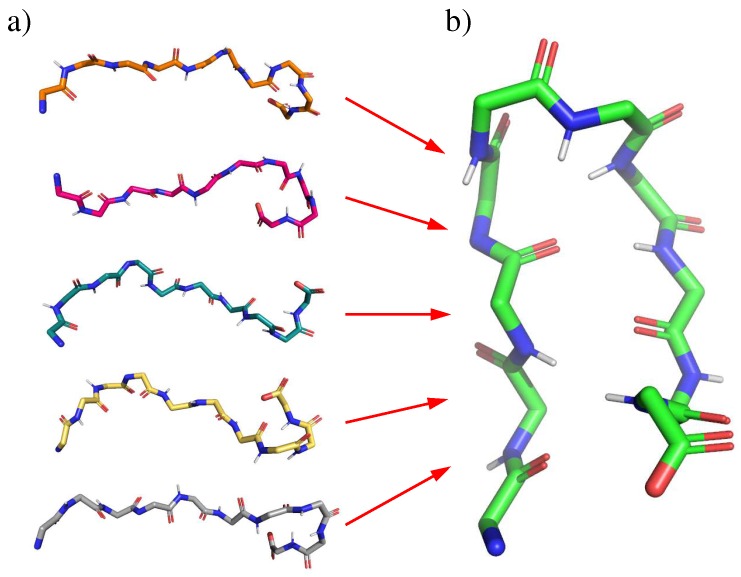
Structural representation of Chignolin. (**a**) A set of extended conformations involving the initial turn at the C-terminal side. (**b**) Folded conformation. Only the protein backbone is represented, using PyMOL [39].

**Figure 3 molecules-24-01150-f003:**
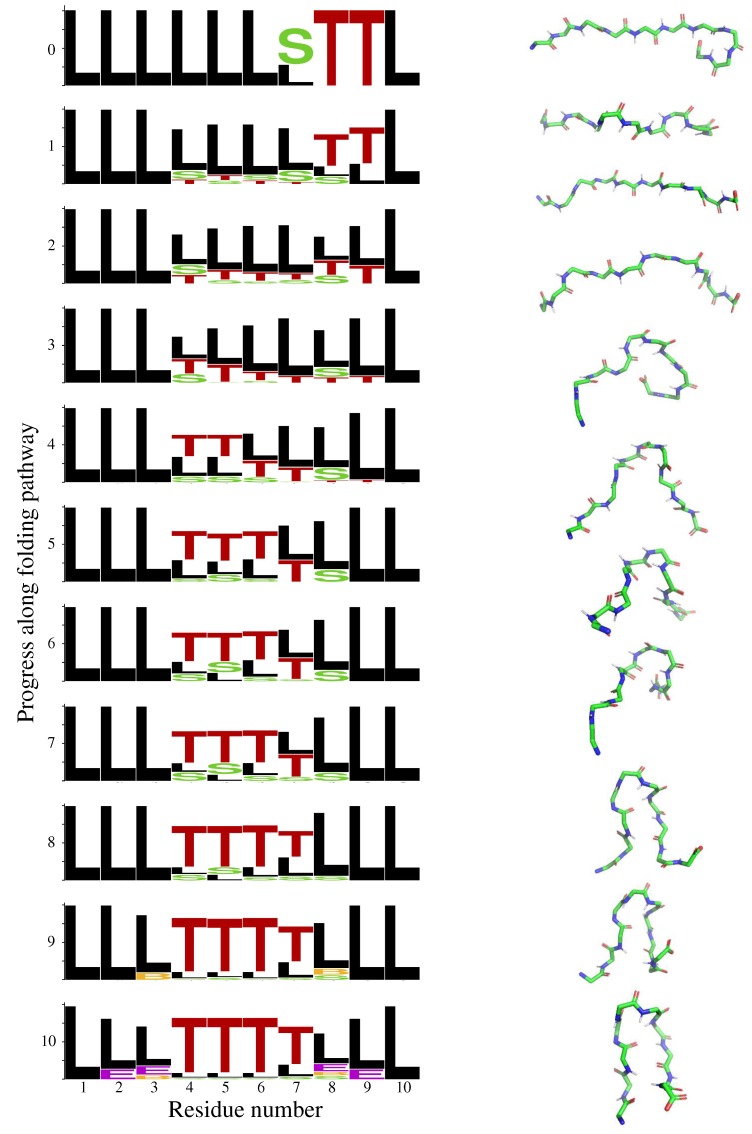
The left side panel represents the evolution of the structural propensities at the residue level along Chignolin folding pathway (see Figure 1 and the associated comments for additional explanations about this representation). The right side panel shows some intermediate states along one of the computed folding paths. Only the protein backbone is represented, using PyMOL [39].

**Figure 4 molecules-24-01150-f004:**
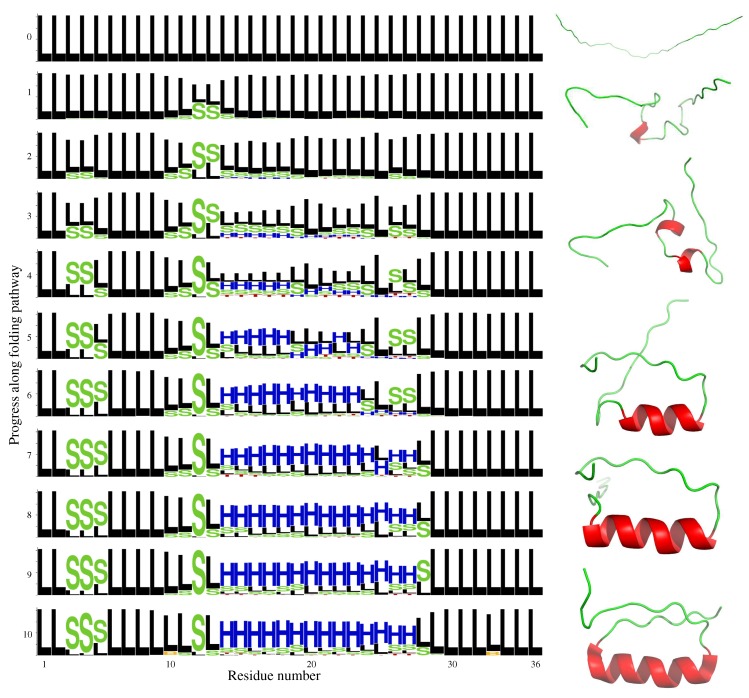
The left side panel represents the evolution of the structural propensities at the residue level along DS119 folding pathway. The right side panel shows some intermediate states along one of the computed folding paths. The “cartoon” representation clearly shows the formation of the helix. PyMOL [39] was used for the structural visualization.

**Figure 5 molecules-24-01150-f005:**
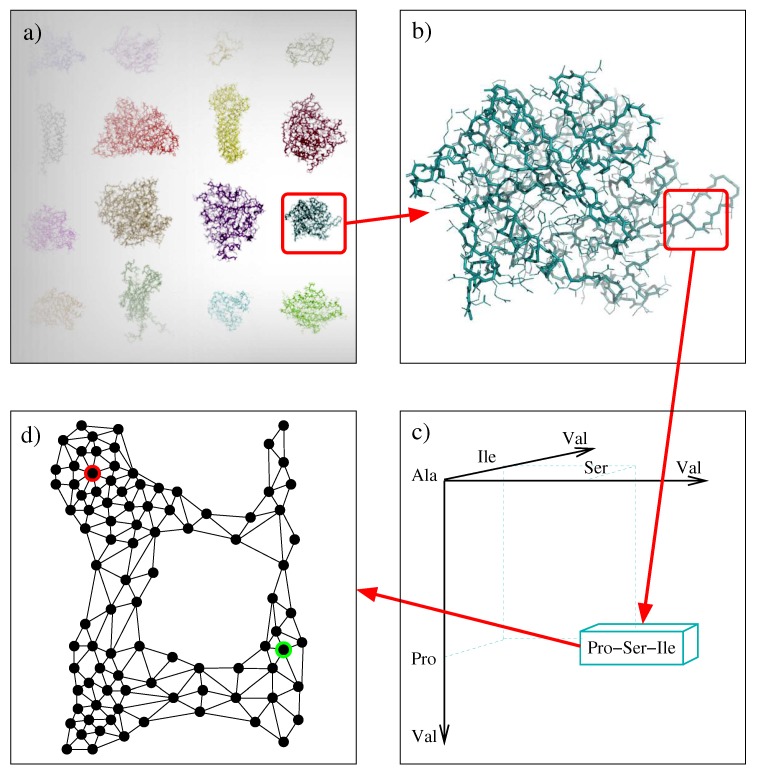
Construction of the tripeptide database: (**a**) A non-redundant set of experimentally-determined protein structures is used as input. (**b**) For each protein, fragments of three consecutive residues (called tripeptides) are analyzed. (**c**) The structural information is stored in a database containing one record for each tripeptide (8000 in total). (**d**) For each tripeptide, the conformations recorded in the database are related with a proximity criterion and structured into an adjacency graph (the figure shows a simplified representation of this graph for tripeptide Pro-Ser-Ile).

**Figure 6 molecules-24-01150-f006:**
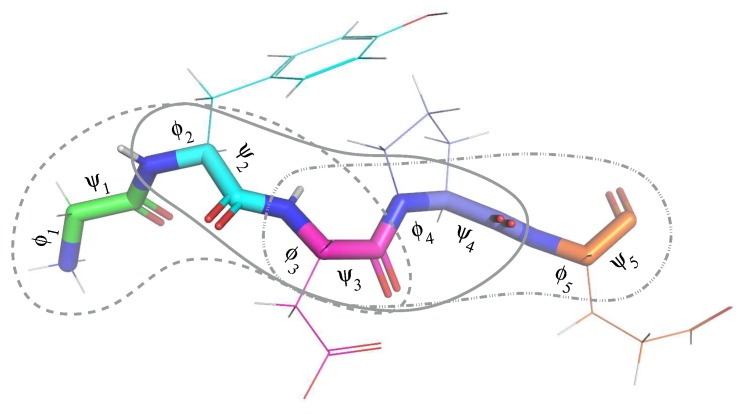
Illustration of a protein fragment involving five residues. Each residue is represented using a different colors for the carbon atoms. The backbone is represented using thicker lines. Considering constant bond lengths, bond angles and peptide bond torsions, the protein backbone conformation can be defined from a pair of angles (ϕ and ψ) for each residue. The gray lines indicate the three overlapping tripeptides composing this five residue fragment.

**Figure 7 molecules-24-01150-f007:**
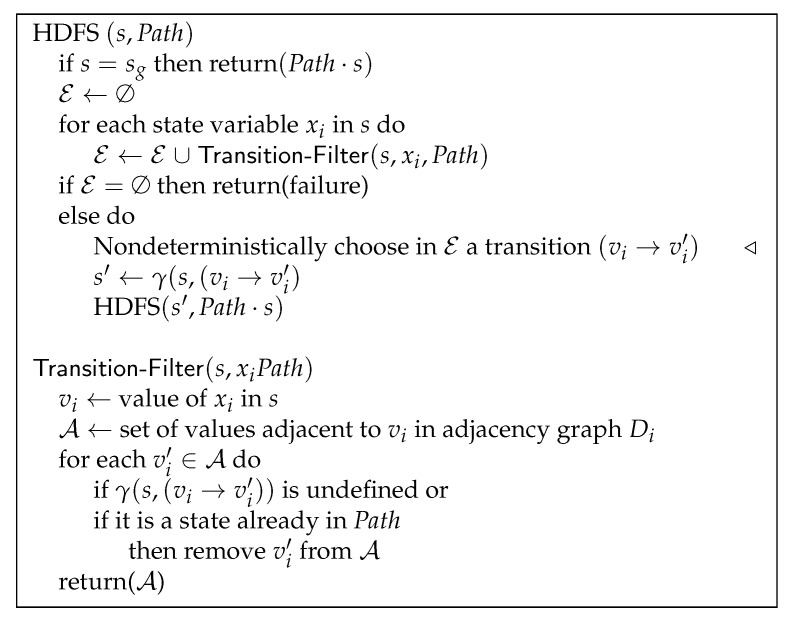
Main procedure as a recursive nondeterministic best-first search. The choice (in step ◃) is guided with the heuristic cost function used to order the set A. In the case of failure, backtracking is performed at this step to other remaining options in the set E, which is computed incrementally.

**Table 1 molecules-24-01150-t001:** Number of conformations (i.e., number of values of state variables) for the eight overlapping tripeptides composing Chignolin.

Tripeptide Sequence	No. of Conformations
Gly-Tyr-Asp	994
Tyr-Asp-Pro	710
Asp-Pro-Glu	1541
Pro-Glu-Thr	1030
Glu-Thr-Gly	1446
Thr-Gly-Thr	1779
Gly-Thr-Trp	545
Thr-Trp-Gly	240

**Table 2 molecules-24-01150-t002:** Performance indicators of the HDFS algorithm to compute different conformational transitions of Chignolin (top) and the mutant Chignolin-W9A (bottom). CPU time was measured with an Intel^®^ Core^TM^ i7 processor at 2.8 GHz, using a single core.

	Chignolin (Original Sequence)			
	C-ter T→folded	C-ter T→misfolded	extended→folded	extended→misfolded
CPU time (s)	11.1	8.7	5.2	3.5
# states	5416.4	2587.7	2800.1	849.5
# backtracks	234.6	136.6	124.6	39.2
Path length (# steps)	133.8	54.5	106.3	48.7
Path distance (rad)	8.8	5.1	6.0	7.0
Path density	**31.9**	5.5	23.3	**4.5**
	**Chignolin-W9A (Mutant)**			
	C-ter T→folded	C-ter T→misfolded	extended→folded	extended→misfolded
CPU time (s)	12.2	8.8	5.6	5.1
# states	4943.6	2567.8	2317.0	2946.0
# backtracks	219.6	139.0	101.3	126.3
Path length (# steps)	140.3	51.3	103.0	125.7
Path distance (rad)	8.2	9.0	5.8	8.2
Path density	**31.2**	4.6	23.4	**23.8**

**Table 3 molecules-24-01150-t003:** Performance indicators of the HDFS algorithm on DS119.

	DS119: Extended→Folded
CPU time (s)	25.2
# states	70558.2
# backtracks	8210.4
Path length (# steps)	158.2
Path distance (rad)	11.3
Path density	124.4

**Table 4 molecules-24-01150-t004:** Number of conformations (i.e., number of values of state variables) for the eight overlapping tripeptides composing DS119.

Tripeptide Sequence	No. Conformations	Tripeptide Sequence	No. Conformations
Gly-Ser-Gly	3727	Lys-Lys-Leu	2286
Ser-Gly-Gln	1118	Lys-Leu-Lys	1996
Gly-Gln-Val	1294	Leu-Lys-Glu	3100
Gln-Val-Arg	607	Leu-Glu-Glu	1631
Val-Arg-Thr	970	Glu-Glu-Ala	2591
Arg-Thr-Ile	757	Glu-Ala-Lys	1514
Thr-Ile-Trp	181	Ala-Lys-Lys	1714
Ile-Trp-Val	180	Lys-Lys-Ala	1629
Trp-Val-Gly	279	Lys-Ala-Asn	1009
Val-Gly-Gly	2443	Ala-Asn-Ile	1010
Gly-Gly-Thr	2510	Asn-Ile-Arg	647
Gly-Thr-Pro	1428	Ile-Arg-Val	998
Thr-Pro-Glu	1738	Arg-Val-Thr	1351
Pro-Glu-Glu	1752	Val-Thr-Phe	888
Glu-Glu-Leu	3433	Thr-Phe-Trp	151
Glu-Leu-Lys	2378	Phe-Trp-Gly	192
Leu-Lys-Lys	2528	Trp-Gly-Asp	257

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
