# Peer review of "Investigating the Formation of Structural Elements in Proteins Using Local Sequence-Dependent Information and a Heuristic Search Algorithm"

_molecules, 2019, doi:10.3390/molecules24061150_

Reviewer 1 Report

In this paper, the authors have developed a search algorithm to underline the folding pathways of proteins. This is an interesting work in which the algorithm relies upon the local structural preferences. They have demonstrated the algorithm using two synthetic protein – Chignolin (& its mutants) and DS119.

However, I do suggest that the authors should elaborate on the relationship between the computational time taken by this heuristic algorithm and the length of the protein.

Author Response

Point 1: In this paper, the authors have developed a search algorithm to underline the folding pathways of proteins. This is an interesting work in which the algorithm relies upon the local structural preferences. They have demonstrated the algorithm using two synthetic protein – Chignolin (& its mutants) and DS119.

However, I do suggest that the authors should elaborate on the relationship between the computational time taken by this heuristic algorithm and the length of the protein.

Response 1: We would like to thank the Reviewer for his/her positive feedback and for suggestion to improve the manuscript, which we implemented in the revised version. In the Methods section, we provide a theoretical analysis of the complexity of the algorithm, which is exponential in the worst case. Nevertheless, in practice, the computing time scales with a reasonable growth with the size of the protein. The empirical scalability analysis using seven proteins, with sizes from 10 to 67 residues, is mentioned in Section 3.3 and detailed as supplementary material. We also explained and insisted that the proposed algorithm, only including local structural information, is not aimed to investigate folding mechanisms of these larger proteins. For this future work, as mentioned in the conclusion, including long-range (non-local) interactions in the heuristic cost function would be required.

Reviewer 2 Report

This manuscript describes a method to predict the folding pathway of a peptide using numeration of possible conformations of tripeptides composing the target peptide by database search developed by the authors themselves. The authors applied their method to relatively short proteins, chignolin and DS119. They demonstrated that their method predicts folding pathways of these proteins and their results are consistent with those of MD simulations. The issue addressed in their manuscript is quite interesting and significant. However, I feel that the following points are reconsidered before publication.

Their method is based on the local structure formation tendency in a protein sequence. The plausibility of the structure depends on density function. I think that the result seems to reflect the final 3D structure of a protein, not an initial folding structure. That is, their results do not necessary reflect folding process. Thus, the authors should discuss and add the detailed justification of their method. In the same context, I think that the authors should add more detailed comparisons of their results with MD simulations and demonstrate the validity of their method. By the way, how were some cartoons in the right panels of Figure 3, and 4 created? The explanations should be added.

I think that their technique may be useful for relatively short proteins, that is, nonlocal contacts (contacts formed by a pair of residues separated on the sequence) are not significant. However, in a larger protein, nonlocal contacts will be significant in the initial stage of folding. For such proteins, even if local fragments (5-6 residues) in two proteins share same sequence, the corresponding segments in the final 3D structures of these proteins may exhibit different 3D structures caused by effects of nonlocal contacts. This means that the folding processes of the corresponding segments in such proteins should differ. I think that the technique of the authors cannot treat this effect. The authors should discuss these points.

Minor point;

In page 9, line 223, there is a sentence “extended fragments at the two ends”. But in Figure 4, L and S are dominant in the both ends, not E. I think that this is confusing. I recommend to rewrite this part.

Author Response

Point 1: This manuscript describes a method to predict the folding pathway of a peptide using numeration of possible conformations of tripeptides composing the target peptide by database search developed by the authors themselves. The authors applied their method to relatively short proteins, chignolin and DS119. They demonstrated that their method predicts folding pathways of these proteins and their results are consistent with those of MD simulations. The issue addressed in their manuscript is quite interesting and significant. However, I feel that the following points are reconsidered before publication.

Response 1: We would like to thank the Reviewer for his/her positive feedback and for suggestions to improve the manuscript, which we implemented in the revised version. Responses to the different questions are provided below.

Point 2: Their method is based on the local structure formation tendency in a protein sequence. The plausibility of the structure depends on density function. I think that the result seems to reflect the final 3D structure of a protein, not an initial folding structure. That is, their results do not necessary reflect folding process. Thus, the authors should discuss and add the detailed justification of their method.

Response 2: We further clarified that the main goal of our work is to investigate the importance of local sequence-dependent structural preferences in the formation (folding mechanism) of structural elements. To test this hypothesis, we developed a heuristic method that computes folding paths giving priority in the search process to “dense” intermediate states. The density function is derived from a structural database of three residue fragments extracted from experimentally-determined protein structures. The database includes fragments extracted from coil regions, which have been shown to be useful elements to model unfolded or disordered proteins. Therefore, we assume that the structural information encoded in the database is not limited to folded states, and that it can be useful to investigate the folding processes. Note that results presented in the manuscript for Chignolin (and several mutants) show that unfolded, folded and intermediate states can be sampled from our database. This has been further clarified in the Section 3.1. In addition, several sentences in the Introduction, which has been reworked, explain the motivation of this work and the considered assumptions.

Point 3: In the same context, I think that the authors should add more detailed comparisons of their results with MD simulations and demonstrate the validity of their method.

Response 3: We have qualitatively compared our results to those described in related literature, showing the capability of our method to identify similar intermediate states along the folding process. However, a more detailed and quantitative comparison is difficult since, to the best of our knowledge, MD trajectories for Chignolin and DS119 are not available. Furthermore, such a detailed comparison would not be very meaningful, since the aims of the two methods, MD and our algorithm, are different: The paths provided by our algorithm are an approximation, from which interesting information about folding mechanisms can already be obtained, but that should be refined (using other methods and models) to get access to accurate information at the atomic level (as provided by MD simulations). On the other hand, our algorithm is orders of magnitude faster than atomistic MD simulations. This discussion has been included in the Conclusion section.

Point 4: By the way, how were some cartoons in the right panels of Figure 3, and 4 created? The explanations should be added.

Response 4: Indeed, we added the required explanation in the figures captions about the use of PyMOL for this purpose.

Point 5: I think that their technique may be useful for relatively short proteins, that is, nonlocal contacts (contacts formed by a pair of residues separated on the sequence) are not significant. However, in a larger protein, nonlocal contacts will be significant in the initial stage of folding. For such proteins, even if local fragments (5-6 residues) in two proteins share same sequence, the corresponding segments in the final 3D structures of these proteins may exhibit different 3D structures caused by effects of nonlocal contacts. This means that the folding processes of the corresponding segments in such proteins should differ. I think that the technique of the authors cannot treat this effect. The authors should discuss these points.

Response 5: The Reviewer is right, and this is further underlined in the revised version of the manuscript: the applicability of the proposed algorithm is limited to small proteins or protein fragments. As mentioned in the Conclusion, extensions of the algorithm including energy/scoring functions to take into account non-local interactions would be an interesting direction for future work. In addition, we insist that the proposed algorithm is not aimed at predicting structures, but at investigating how these structures form starting from unstructured states.  Therefore, the type of system mentioned by the Reviewer (same sequence that folds into different 3D structures) could be investigated using our method, and indeed, could be an interesting case study. For such type of system, it is possible that the database contains enrichment in different structural classes for the same tripeptide, and that a series of consecutive tripeptides have no special structural prevalence, but one of them can be eventually stabilized in the context of long-range interactions.

Point 6: Minor point;

In page 9, line 223, there is a sentence “extended fragments at the two ends”. But in Figure 4, L and S are dominant in the both ends, not E. I think that this is confusing. I recommend to rewrite this part.

Response 6: We agree that the use of the word “extended” can be misleading. Actually, the E code in the output of DSSP stands for an extended strand in parallel and/or anti-parallel beta-sheet conformation. Extended regions non involved in beta-sheets are not labeled E by DSSP; they appear as L in our figures. This can be clearly seen for the initial state of DSS119 in Figure 4. In the text, we have replaced “extended” by “relatively extended”.  

Reviewer 3 Report

Estaña et al. describe an algorithm to study the folding of proteins using a discrete, fragment-based representation. This research direction is very interesting, and I hope that it will bear fruit. However, the current manuscript claims makes a very bold claim, namely: ”The simplicity and

speed of the proposed algorithm will allow the application to study the folding of larger biomolecules and to explore partner recognition events.”

In my opinion, this claim must be supported either on theoretical grounds (using a detailed discussion on thermodynamics and free energy) or on computational grounds (by demonstrating, convincingly and quantitatively, that the algorithm gives results equivalent to MD simulation). In its current state, the manuscript does neither, as outlined below. Unless this claim can be supported one way or the other, it must be reformulated such as to make clear that more research is needed first.

From a theoretical perspective, an in-depth discussion of the free energy of folding is required, which is currently missing. There is a “cost function” which looks to be the energy function, but it is only defined in the Materials and Methods, and is not discussed in the main text. In particular:

- The fragment library is derived only from folded structures. It should be argued why the same library can be applied to unfolded structures.

- The fragment energy rho seems to be only based on the connectivity graph. Why not use the actual count of each fragment in the database as well?

- It is assumed that every fragment can be followed by any other fragment as long as the angles are similar enough (equation 2). This neglects other statistical knowledge about local structure in proteins, in particular the favorable n-n+4 hydrogen bond interaction that defines the alpha helix. This neglect must be justified in the manuscript.

- Non-bonded energies are modeled in a peculiar way. Instead of adding them to the energy function, the authors employ an algorithm that avoids collisions between pseudo-atoms. This neglects favorable interactions, in particular hydrophobic interactions and strand-strand hydrogen bonds. This neglect must be justified in the manuscript.

Alternatively, from a computational perspective, the following points are missing:

- The section where the results are compared to that of MD simulations is embarassingly vague. The claim that the results show that they are similar MUST be supported by pairwise RMSD calculations over the entire trajectory, showing that they explore the same energy minima in roughly the same order.

- If statistical knowledge of local structure and non-bonded interactions are neglected in the explicit energy evaluation, it should be demonstrated that this neglect has no practical consequences. In other words, the sampled models should have correct fragment transition frequencies and correct non-bonded geometry, and any deviations should be quantified.

- For the collision detection, the radius of each pseudo-atom should be mentioned.

In addition, I have the following general points:

- There are no results pertaining to partner recognition events. This part of the claim must be removed no matter what.

- The language of this paper is often confusing, and seems more suited for a computer science journal. Concepts such as “heuristic”, “depth-first”, “admissible” and “complete” are assumed to be known, and familiarity with dynamic programming (Dijkstra’s algorithm) is assumed, which seems to me optimistic for a chemistry readership. In contrast, space is wasted in defining trivial concepts such as tripeptides, rotamers and residues.

- Figure 1d is somewhat misleading, since the structures are explicitly driven towards the known folded state (using parameter w1). What would be more relevant is the closest approximation of any structure of the fragment library. This can be computed using Kolodny’s greedy algorithm.

- It is not clear to me how the authors “non-deterministically choose” a transition. Are all transitions sampled exhaustively, or is one chosen at random? In case of the former, I do not understand how the algorithm would be heuristic. In case of the latter, I do not understand how it could be complete, as is claimed.

- I would like to see a better estimate of how large the search space is, and much of it is being explored. Currently, only a naive upper bound is mentioned, whereas for any given protein sequence, the total search space can be exactly computed using dynamic programming. I am also curious how much of the search space is reachable from the folded structure, which could perhaps be estimated by running the algorithm in reverse.

Author Response

Point 1: Estaña et al. describe an algorithm to study the folding of proteins using a discrete, fragment-based representation. This research direction is very interesting, and I hope that it will bear fruit.

Response 1: We would like to thank the Reviewer for his/her positive feedback and for encouraging us to continue on this research direction. We have taken into account his/her remarks and suggestion to further clarify and improve the revised version of the paper. Responses to the different questions are provided below.

Point 2: However, the current manuscript claims makes a very bold claim, namely: ”The simplicity and speed of the proposed algorithm will allow the application to study the folding of larger biomolecules and to explore partner recognition events.” In my opinion, this claim must be supported either on theoretical grounds (using a detailed discussion on thermodynamics and free energy) or on computational grounds (by demonstrating, convincingly and quantitatively, that the algorithm gives results equivalent to MD simulation). In its current state, the manuscript does neither, as outlined below. Unless this claim can be supported one way or the other, it must be reformulated such as to make clear that more research is needed first.

Response 2:  We agree with the Reviewer that the mentioned sentence can be misleading. We reformulated it to be in accordance with the contribution, clarifying in conclusion what we intend to pursue in future work.

Point 3: From a theoretical perspective, an in-depth discussion of the free energy of folding is required, which is currently missing.

Response 3: The Introduction of the manuscript provides references to papers about protein folding/unfolding where the interested reader could find explanations and discussions about folding free energy and energy landscapes, issues about which we do not have any specific contribution. However, the Reviewer is right that some discussion relating the notion of density used in the heuristic with the notion of free energy was missing in the original submission. We have made changes in the revised version to clarify  this point.

Point 4: There is a “cost function” which looks to be the energy function, but it is only defined in the Materials and Methods, and is not discussed in the main text.

Response 4: The cost function is defined in the main text (Section 3.3). It is used to heuristically guide the search, favoring transitions to neighboring states with higher density and possibly leading to a shorter path towards the target state. This is further detailed in the revised manuscript. The introduction of the paper only explains the intuition behind this cost function since the required material to explain the equation in detail and the use of this cost function in our algorithm have not been introduced at that point.

Point 5: In particular:

- The fragment library is derived only from folded structures. It should be argued why the same library can be applied to unfolded structures.

Response 5: The structural database of three residue fragments was constructed from experimentally-determined protein structures, including fragments extracted from coil regions. Several works (including our recent work, ref. 23 in the manuscript) have shown that structural libraries built from coil regions are particularly useful to model unfolded or disordered proteins. Therefore, we assume that the structural information encoded in our exhaustive database is not limited to folded states, and that it can be useful to investigate the folding processes. This point has been further clarified in Section 3.1.

Point 6: - The fragment energy rho seems to be only based on the connectivity graph. Why not use the actual count of each fragment in the database as well?

Response 6: rho is a local measure used within the heuristic to select the next state of a tripeptide being visited during the search. Since the heuristic cost function is applied at the single tripeptide level (i.e. at the state variable level), a “normalization” using the number of conformations contained in the database is not necessary.  

Point 7: - It is assumed that every fragment can be followed by any other fragment as long as the angles are similar enough (equation 2). This neglects other statistical knowledge about local structure in proteins, in particular the favorable n-n+4 hydrogen bond interaction that defines the alpha helix. This neglect must be justified in the manuscript.

Response 7: This important issue is further clarified in the revised manuscript, and details are provided as supplementary material. As underlined in the Introduction, the main goal of our work is to investigate the importance of local sequence-dependent structural preferences in the formation (folding mechanism) of structural elements. Therefore, other effects such as hydrogen bonds in helices are neglected (in an explicit way) within the path search algorithm. However, during the path search process, the structural preferences of a tripeptide tend to be propagated to the neighbours, due to the constraints imposed on the angles. Indeed, the distribution of the phi-psi angles for a residue does not depend only on the nature of the neighboring residues, but also on their structure. This is illustrated below in Figure 1. This natural “cooperativity”, which somehow incorporates (in an implicit way) the effect of n-n+4 interactions within helices, is encoded in the tripeptide database. We have observed this effect in our recent work on IDPs modeling also using a tripeptide database (ref. 23 in the manuscript):  long helical regions are propagated due to the structural effects exerted by the neighbouring tripeptides, without explicitly considering hydrogen-bonds.

Figure 1 : Distributions of the phi-psi angles of the middle residue in a tripeptide, Ser-Arg-Ala, depending on the structure of the neighboring residues. a) All the values for the central residue Arg, independently on the structure of Ser and Ala. b)  Values for Arg when Ser and Ala are in the alpha region. c) Values for Arg when Ser and Ala are in the beta/polyproline-II region.

Point 8: - Non-bonded energies are modeled in a peculiar way. Instead of adding them to the energy function, the authors employ an algorithm that avoids collisions between pseudo-atoms. This neglects favorable interactions, in particular hydrophobic interactions and strand-strand hydrogen bonds. This neglect must be justified in the manuscript.

Response 8: As further clarified in the paper (and in the answer above), we do not consider energies in the heuristic on purpose, with the aim to investigate the importance of local structural preferences encoded in the tripeptide database. We only filter out conformations involving collisions during the path search, since they are unrealistic. The Conclusion underlines for future work that,  including an energy function taking into account interactions between distant pairs or residues (in sequence) can be important to investigate larger molecules, or aspects related to the stability. However, the results presented in the paper illustrate the importance of local, sequence-dependent structural preferences in the formation of  structural elements in proteins.

Point 9: Alternatively, from a computational perspective, the following points are missing:

- The section where the results are compared to that of MD simulations is embarassingly vague. The claim that the results show that they are similar MUST be supported by pairwise RMSD calculations over the entire trajectory, showing that they explore the same energy minima in roughly the same order.

Response 9: In this work, we have qualitatively compared our results to those described in related literature, showing the capability of our method to identify similar intermediate states along the folding process. However, a more detailed and quantitative comparison is difficult since, to the best of our knowledge, MD trajectories for Chignolin and DS119 are not available. Furthermore, such a detailed comparison would not be very meaningful, since the aims of the two methods, MD and our algorithm, are different: The paths provided by our algorithm are only approximations, from which interesting information about folding mechanisms can already be obtained, but that should be refined (using other methods and models) to get access to accurate information at the atomic level (as provided by MD simulations). On the other hand, our algorithm is orders of magnitude faster than atomistic MD simulations. These issues are further underlined in the paper.

Point 10: - If statistical knowledge of local structure and non-bonded interactions are neglected in the explicit energy evaluation, it should be demonstrated that this neglect has no practical consequences. In other words, the sampled models should have correct fragment transition frequencies and correct non-bonded geometry, and any deviations should be quantified.

Response 10: Actually, we do not neglect statistical knowledge of local structure, since this is the basis of our approach. We neglect (explicitly) non-bonded interactions, but for the interactions between neighbor residues implicitly included in the database. As mentioned above, there is a propagation mechanism that incorporates interactions along the sequence. Therefore, only long-distance interactions across the protein are totally neglected, excepting for the collision test. This has been clarified in the revised version of the manuscript.

Point 11: - For the collision detection, the radius of each pseudo-atom should be mentioned.

Response 11: The Reviewer is right: the required explanations have been included in the revised version of the manuscript. For the sidechain pseudo-atoms, we used the values proposed by Levitt (J. Mol. Biol., 1976). For the backbone atoms, we used Van der Waals radii proposed by Bondi (J. Phys. Chem., 1964). A pair of non-bonded (pseudo-)atoms was considered to be in collision if their distance was less than 65% of the sum of their radii.

Point 12: In addition, I have the following general points:

- There are no results pertaining to partner recognition events. This part of the claim must be removed no matter what.

Response 12: The last part of the abstract has been reworded. The current contribution does not show results related to partner recognition, but we believe that this is an interesting direction for a followup future work, particularly in the context of intrinsically disordered proteins. 

Point 13: - The language of this paper is often confusing, and seems more suited for a computer science journal. Concepts such as “heuristic”, “depth-first”, “admissible” and “complete” are assumed to be known, and familiarity with dynamic programming (Dijkstra’s algorithm) is assumed, which seems to me optimistic for a chemistry readership. In contrast, space is wasted in defining trivial concepts such as tripeptides, rotamers and residues.

Response 13: The Reviewer is right. We clarified in Section 3.3 the algorithmic concepts and removed notions that are not needed for understanding the proposed computational approach. On the other hand, since we expect that this manuscript will be of interest for computer scientists, we prefer to keep some definitions of basic concepts in structural biology.

Point 14: - Figure 1d is somewhat misleading, since the structures are explicitly driven towards the known folded state (using parameter w1). What would be more relevant is the closest approximation of any structure of the fragment library. This can be computed using Kolodny’s greedy algorithm.

Response 14: The structure represented in orange in Figures 1d is the closest one to the experimentally determined structure of Chignolin obtained from a small set of 1000 conformations sampled from the tripeptide database (this has been clarified in the figure caption). The goal of this test was to show that unfolded, folded and intermediate states can be sampled from the database. For the path search process, indeed, we use the closest state built from the database to the known folded state. Since states are defined from bond torsion angles, we do not need to apply Kolodny’s greedy algorithm to search for the closest one. A simple search process from angle values contained in the database is enough. For Chignolin, the angular RMSD between the experimental structure and the closest state built from the database is 0.1 rad. For DS119,  the angular RMSD is 0.06 rad. This information is now provided in the revised version of the manuscript.

Point 15: - It is not clear to me how the authors “non-deterministically choose” a transition. Are all transitions sampled exhaustively, or is one chosen at random? In case of the former, I do not understand how the algorithm would be heuristic. In case of the latter, I do not understand how it could be complete, as is claimed.

Response 15: This point was clarified in the text of Section 3.3 and caption of the algorithm. The ‘nondeterministic choice’ is a classical convenient algorithmic notation in a pseudo-code for designating a branching point, with backtrack in case of failure over all possible options, in our case the set E. This choice is heuristically guided with the cost function used to order the set A in the function ‘Transition-Filter’. The set E is computed incrementally when needed. There is no sampling. If needed, all options in the set E are explored, which explain the theoretical completeness. In practice, the heuristic guidance makes the search well focused  as shown in the supplementary material.

Point 16: - I would like to see a better estimate of how large the search space is, and much of it is being explored. Currently, only a naive upper bound is mentioned, whereas for any given protein sequence, the total search space can be exactly computed using dynamic programming. I am also curious how much of the search space is reachable from the folded structure, which could perhaps be estimated by running the algorithm in reverse.

Response 16: The Reviewer is right: the total size of search space for each given problem can be computed, taking into account dependencies between variables, using dynamic programming. This is mentioned in the text, but we explained that such information is not very useful in practice. Indeed, it is not the case that much of this total space is explored; only a very small fraction of it is actually visited. The supplementary material details from empirical evidence for proteins ranging from 10 to 67 residues the size of the search space explored. It shows a scalable growth of this search space for a reasonable range of proteins length.

Round  2

Reviewer 2 Report

I tnink that the authors revised thier manuscript adequately. Therefore, I feel that this manuscript can be published in molecules.

Author Response

I think that the authors revised their manuscript adequately. Therefore, I feel that this manuscript can be published in molecules.

Response: We would like to thank again the Reviewer for his/her valuable feedback to improve the manuscript.

Reviewer 3 Report

The revised version of the manuscript has much improved. The abstract now makes now an appropriate claim about the merit of the work, which is now much better explained.

I have only one major issue, which is the claim in the first sentence of the conclusion: "Despite the simplicity of both the algorithm and the heuristic, the results presented in this paper show that the proposed approach identifies relevant protein folding pathways.". This claim remains unsupported by the data, and I insist that it must be changed, else the manuscript is not acceptable.

A fine substitute would be the last sentence of the abstract: “[The results presented in this paper show that] the proposed approach, thanks to its simplicity and computational efficiency, is a promising research direction”. Alternatively, the claim could be diluted, e.g. "Despite the simplicity of both the algorithm and the heuristic, the results presented in this paper show that the proposed approach constitutes a promising new research direction towards the identification of relevant protein folding pathways."

In addition, I am a bit worried about the scalability analysis. While it is true that at 64 residues and 1 CPU hour, the algorithm is still faster than explicit-solvent MD (although I am not sure about “multiple orders of magnitude”), the scaling coefficient alpha=3.4 is rather high, and rather at odds with the linear scaling, i.e. alpha=1, reported in the paper. Naive MD scales quadratically (alpha=2) for a single time step, although the appropriate number of time steps typically also increases with system size. Modern MD implementations have alpha much smaller than 2. This implies that MD, while much slower for smaller systems, scales better than the proposed method. This seems to me an important avenue for future research.

For the rest, I have objections to some of the responses, as outlined below, but if the authors judge otherwise, I will not object.

Point 6: - The fragment energy rho seems to be only based on the connectivity graph. Why not use the actual count of each fragment in the database as well?

Response 6: rho is a local measure used within the heuristic to select the next state of a tripeptide being visited during the search. Since the heuristic cost function is applied at the single tripeptide level (i.e. at the state variable level), a “normalization” using the number of conformations contained in the database is not necessary.

Objection: I don’t see how this could be true. Imagine that all protein structures consist of only two tripeptide conformations, A and B, that succeed each other randomly (i.e they are connected in the compatibility graph). Imagine a database prevalence of 90 % A, 10 % B. How is this prevalence reproduced in the sampling using the author’s framework? Perhaps I am interpreting the cost/transition formulae wrongly, but then, perhaps some of the readers could make the same mistake as I do. I recommend some clarification.

Point 9:

The section where the results are compared to that of MD simulations is embarassingly vague. The claim that the results show that they are similar MUST be supported by pairwise RMSD calculations over the entire trajectory, showing that they explore the same energy minima in roughly the same order.

Response 9: In this work, we have qualitatively compared our results to those described in related literature, showing the capability of our method to identify similar intermediate states along the folding process. However, a more detailed and quantitative comparison is difficult since, to the best of our knowledge, MD trajectories for Chignolin and DS119 are not available. Furthermore, such a detailed comparison would not be very meaningful, since the aims of the two methods, MD and our algorithm, are different: The paths provided by our algorithm are only approximations, from which interesting information about folding mechanisms can already be obtained, but that should be refined (using other methods and models) to get access to accurate information at the atomic level (as provided by MD simulations). On the other hand, our algorithm is orders of magnitude faster than atomistic MD simulations. These issues are further underlined in the paper.

Objection: The aims of the two methods, MD and the presented algorithm, seem not at all different to me. This impression is reinforced by the statement (repeated twice in various wordings)  “[…] show the interest of the proposed approach, which is extremely fast when compared with currently-used computational methods based on molecular dynamics (MD) simulations”. I am all in favour of replacing MD with something more efficient, but it must be shown that the accuracy will not suffer too much.

The statement “quantitative comparison is difficult since, to the best of our knowledge, MD trajectories for Chignolin and DS119 are not available” does not add to the credibility of the paper. “Arguments from ignorance” are a logical fallacy that cannot be used to support a claim. In any case, to the best of my knowledge, chignolin is widely used as a tutorial system to teach MD simulation, and trajectories shouldn’t be too hard to obtain. Fast, open-source MD implementations such as Gromacs can be used to generate chignolin MD trajectories using little computational resources.

Point 10: - If statistical knowledge of local structure and non-bonded interactions are neglected in the explicit energy evaluation, it should be demonstrated that this neglect has no practical consequences. In other words, the sampled models should have correct fragment transition frequencies and correct non-bonded geometry, and any deviations should be quantified.

Response 10: Actually, we do not neglect statistical knowledge of local structure, since this is the basis of our approach. We neglect (explicitly) non-bonded interactions, but for the interactions between neighbor residues implicitly included in the database. As mentioned above, there is a propagation mechanism that incorporates interactions along the sequence. Therefore, only long-distance interactions across the protein are totally neglected, excepting for the collision test. This has been clarified in the revised version of the manuscript.

Objection: I believe there is some confusion about the word “local”. Algorithms for fold detection and secondary structure prediction are based on local structure, in the sense that they consider the statistical probability of structural fragments *and* their transition frequencies. This can be modelled either as a Markov chain (fragments are states) or using a Hidden Markov Model (fragments are emitted by states). The authors’ approach does not model transitions apart from what  is defined as “feasible” in terms of angles (equation 2) and lack of clashes. Statistical knowledge on known proteins (e.g. the histogram of observed helix lengths in the PDB) is neglected. This neglect is only justified if the correct transition frequencies are emergent from the approach, and this must be demonstrated  (at least, if any claims about finding relevant folding intermediates are being made).

Point 14: - Figure 1d is somewhat misleading, since the structures are explicitly driven towards the known folded state (using parameter w1). What would be more relevant is the closest approximation of any structure of the fragment library. This can be computed using Kolodny’s greedy algorithm.

Response 14: The structure represented in orange in Figures 1d is the closest one to the experimentally determined structure of Chignolin obtained from a small set of 1000 conformations sampled from the tripeptide database (this has been clarified in the figure caption). The goal of this test was to show that unfolded, folded and intermediate states can be sampled from the database. For the path search process, indeed, we use the closest state built from the database to the known folded state. Since states are defined from bond torsion angles, we do not need to apply Kolodny’s greedy algorithm to search for the closest one. A simple search process from angle values contained in the database is enough. For Chignolin, the angular RMSD between the experimental structure and the closest state built from the database is 0.1 rad. For DS119, the angular RMSD is 0.06 rad. This information is now provided in the revised version of the manuscript.

Objection: The authors are correct, if angular RMSD is used as a measurement of similarity. This is more-or-less reasonable for small peptides (such as chignolin). However, for larger molecules, it is the absolute coordinate frame that must be taken into account. Small angular deviations can accumulate into large coordinate frame deviations. Conversely, very different angles may combine into very similar coordinate frame transformations. As the latter is the basis of loop closure algorithms, of which the authors know much more than I do, I am surprised that they favor angular RMSD rather than Cartesian RMSD (which is used in Kolodny’s greedy algorithm).

Point 15: - It is not clear to me how the authors “non-deterministically choose” a transition. Are all transitions sampled exhaustively, or is one chosen at random? In case of the former, I do not understand how the algorithm would be heuristic. In case of the latter, I do not understand how it could be complete, as is claimed.

Response 15: This point was clarified in the text of Section 3.3 and caption of the algorithm. The ‘nondeterministic choice’ is a classical convenient algorithmic notation in a pseudo-code for designating a branching point, with backtrack in case of failure over all possible options, in our case the set E. This choice is heuristically guided with the cost function used to order the set A in the function ‘Transition-Filter’. The set E is computed incrementally when needed. There is no sampling. If needed, all options in the set E are explored, which explain the theoretical completeness. In practice, the heuristic guidance makes the search well focused as shown in the supplementary material.

Objection: It is not stated if the algorithm terminates A) as soon as the *first* valid path is found, or B) that it continues until *all* paths have been found. I could not find any clarification in the supplementary material. In case B), the algorithm would be complete *for a given random starting state* (and the heuristics would just be an implementation detail). Even then, I see no proof why all states that can reach the termination state would be visited (which I feel is what “completeness” implies).

Author Response

Point 1: The revised version of the manuscript has much improved. The abstract now makes now an appropriate claim about the merit of the work, which is now much better explained.

Response 1: We would like to thank the Reviewer for his/her careful review and feedback, which was very valuable, helping us to improve the manuscript.

Point 2: I have only one major issue, which is the claim in the first sentence of the conclusion: "Despite the simplicity of both the algorithm and the heuristic, the results presented in this paper show that the proposed approach identifies relevant protein folding pathways.". This claim remains unsupported by the data, and I insist that it must be changed, else the manuscript is not acceptable.

A fine substitute would be the last sentence of the abstract: “[The results presented in this paper show that] the proposed approach, thanks to its simplicity and computational efficiency, is a promising research direction”. Alternatively, the claim could be diluted, e.g. "Despite the simplicity of both the algorithm and the heuristic, the results presented in this paper show that the proposed approach constitutes a promising new research direction towards the identification of relevant protein folding pathways."

Response 2: We agree with the Reviewer. We have changed this sentence as suggested.  

Point 3: In addition, I am a bit worried about the scalability analysis. While it is true that at 64 residues and 1 CPU hour, the algorithm is still faster than explicit-solvent MD (although I am not sure about “multiple orders of magnitude”), the scaling coefficient alpha=3.4 is rather high, and rather at odds with the linear scaling, i.e. alpha=1, reported in the paper. Naive MD scales quadratically (alpha=2) for a single time step, although the appropriate number of time steps typically also increases with system size. Modern MD implementations have alpha much smaller than 2. This implies that MD, while much slower for smaller systems, scales better than the proposed method. This seems to me an important avenue for future research.

Response 3: There is a confusion between the complexity of a single simulation step, which is  in O(n^2) for MD, but is constant in our case, and the complexity of the entire search, which would require for both approaches a large number of such steps. This confusion reveals that our text was not clear enough on this point. In the revised version, we clarified the issue (section 3.3) and we developed it in detail in the supplementary S1 section.  

For the rest, I have objections to some of the responses, as outlined below, but if the authors judge otherwise, I will not object.

Point 6: - The fragment energy rho seems to be only based on the connectivity graph. Why not use the actual count of each fragment in the database as well?

Response 6: rho is a local measure used within the heuristic to select the next state of a tripeptide being visited during the search. Since the heuristic cost function is applied at the single tripeptide level (i.e. at the state variable level), a “normalization” using the number of conformations contained in the database is not necessary.

Objection: I don’t see how this could be true. Imagine that all protein structures consist of only two tripeptide conformations, A and B, that succeed each other randomly (i.e they are connected in the compatibility graph). Imagine a database prevalence of 90 % A, 10 % B. How is this prevalence reproduced in the sampling using the author’s framework? Perhaps I am interpreting the cost/transition formulae wrongly, but then, perhaps some of the readers could make the same mistake as I do. I recommend some clarification.

Response: It seems that the Reviewer misunderstood part of the approach, probably due to the terminology. The transitions are computed between connected states at the tripeptide level. The density is computed for each state individually, not for clusters of states that could correspond to “conformations”. Therefore, the cost function tends to favor transition paths going through dense regions in the state-space, without any need of weighting or normalization.  In the hypothetical case mentioned, the only possible transitions that would be tested are from A to B and from B to A and no sampling would be performed.

Point 9:

The section where the results are compared to that of MD simulations is embarassingly vague. The claim that the results show that they are similar MUST be supported by pairwise RMSD calculations over the entire trajectory, showing that they explore the same energy minima in roughly the same order.

Response 9: In this work, we have qualitatively compared our results to those described in related literature, showing the capability of our method to identify similar intermediate states along the folding process. However, a more detailed and quantitative comparison is difficult since, to the best of our knowledge, MD trajectories for Chignolin and DS119 are not available. Furthermore, such a detailed comparison would not be very meaningful, since the aims of the two methods, MD and our algorithm, are different: The paths provided by our algorithm are only approximations, from which interesting information about folding mechanisms can already be obtained, but that should be refined (using other methods and models) to get access to accurate information at the atomic level (as provided by MD simulations). On the other hand, our algorithm is orders of magnitude faster than atomistic MD simulations. These issues are further underlined in the paper.

Objection: The aims of the two methods, MD and the presented algorithm, seem not at all different to me. This impression is reinforced by the statement (repeated twice in various wordings)  “[…] show the interest of the proposed approach, which is extremely fast when compared with currently-used computational methods based on molecular dynamics (MD) simulations”. I am all in favour of replacing MD with something more efficient, but it must be shown that the accuracy will not suffer too much.

The statement “quantitative comparison is difficult since, to the best of our knowledge, MD trajectories for Chignolin and DS119 are not available” does not add to the credibility of the paper. “Arguments from ignorance” are a logical fallacy that cannot be used to support a claim. In any case, to the best of my knowledge, chignolin is widely used as a tutorial system to teach MD simulation, and trajectories shouldn’t be too hard to obtain. Fast, open-source MD implementations such as Gromacs can be used to generate chignolin MD trajectories using little computational resources.

Response: MD simulations are mentioned several times in the manuscript since they are the most widely used method to understand protein motions, thus being a reference for most readers. Indeed, MD simulations can be very useful to understand the temporal evolution of the conformation of a protein, but are computationally expensive. Although our method is not as accurate as atomistic MD, it can provide information about folding mechanisms in a fast manner and using very small computing resources. These two approaches (MD and our method) are complementary, and should not be placed at the same level. We have rewritten several sentences in the manuscript taking into account the Reviewer’s comment.  

Point 10: - If statistical knowledge of local structure and non-bonded interactions are neglected in the explicit energy evaluation, it should be demonstrated that this neglect has no practical consequences. In other words, the sampled models should have correct fragment transition frequencies and correct non-bonded geometry, and any deviations should be quantified.

Response 10: Actually, we do not neglect statistical knowledge of local structure, since this is the basis of our approach. We neglect (explicitly) non-bonded interactions, but for the interactions between neighbor residues implicitly included in the database. As mentioned above, there is a propagation mechanism that incorporates interactions along the sequence. Therefore, only long-distance interactions across the protein are totally neglected, excepting for the collision test. This has been clarified in the revised version of the manuscript.

Objection: I believe there is some confusion about the word “local”. Algorithms for fold detection and secondary structure prediction are based on local structure, in the sense that they consider the statistical probability of structural fragments *and* their transition frequencies. This can be modelled either as a Markov chain (fragments are states) or using a Hidden Markov Model (fragments are emitted by states). The authors’ approach does not model transitions apart from what  is defined as “feasible” in terms of angles (equation 2) and lack of clashes. Statistical knowledge on known proteins (e.g. the histogram of observed helix lengths in the PDB) is neglected. This neglect is only justified if the correct transition frequencies are emergent from the approach, and this must be demonstrated  (at least, if any claims about finding relevant folding intermediates are being made).

Response: Our approach is different from other fragment-based methods in the way that structural information is encoded. Nevertheless, as explained in the manuscript and in our previous response, structural propensities are taken into account. We believe that this point is clearly explained, and we do not see how we can improve its description in the manuscript.

Point 14: - Figure 1d is somewhat misleading, since the structures are explicitly driven towards the known folded state (using parameter w1). What would be more relevant is the closest approximation of any structure of the fragment library. This can be computed using Kolodny’s greedy algorithm.

Response 14: The structure represented in orange in Figures 1d is the closest one to the experimentally determined structure of Chignolin obtained from a small set of 1000 conformations sampled from the tripeptide database (this has been clarified in the figure caption). The goal of this test was to show that unfolded, folded and intermediate states can be sampled from the database. For the path search process, indeed, we use the closest state built from the database to the known folded state. Since states are defined from bond torsion angles, we do not need to apply Kolodny’s greedy algorithm to search for the closest one. A simple search process from angle values contained in the database is enough. For Chignolin, the angular RMSD between the experimental structure and the closest state built from the database is 0.1 rad. For DS119, the angular RMSD is 0.06 rad. This information is now provided in the revised version of the manuscript.

Objection: The authors are correct, if angular RMSD is used as a measurement of similarity. This is more-or-less reasonable for small peptides (such as chignolin). However, for larger molecules, it is the absolute coordinate frame that must be taken into account. Small angular deviations can accumulate into large coordinate frame deviations. Conversely, very different angles may combine into very similar coordinate frame transformations. As the latter is the basis of loop closure algorithms, of which the authors know much more than I do, I am surprised that they favor angular RMSD rather than Cartesian RMSD (which is used in Kolodny’s greedy algorithm).

Response: We agree with the reviewer that Cartesian RMSD would be a better choice for larger molecules. However, numerous works suggest that angular RMSD is a better metric to compare conformations of small peptides or protein fragments. In addition, angular RMSD does not require structural alignment.  For these reasons, we decided to use angular RMSD in this work.

Point 15: - It is not clear to me how the authors “non-deterministically choose” a transition. Are all transitions sampled exhaustively, or is one chosen at random? In case of the former, I do not understand how the algorithm would be heuristic. In case of the latter, I do not understand how it could be complete, as is claimed.

Response 15: This point was clarified in the text of Section 3.3 and caption of the algorithm. The ‘nondeterministic choice’ is a classical convenient algorithmic notation in a pseudo-code for designating a branching point, with backtrack in case of failure over all possible options, in our case the set E. This choice is heuristically guided with the cost function used to order the set A in the function ‘Transition-Filter’. The set E is computed incrementally when needed. There is no sampling. If needed, all options in the set E are explored, which explain the theoretical completeness. In practice, the heuristic guidance makes the search well focused as shown in the supplementary material.

Objection: It is not stated if the algorithm terminates A) as soon as the *first* valid path is found, or B) that it continues until *all* paths have been found. I could not find any clarification in the supplementary material. In case B), the algorithm would be complete *for a given random starting state* (and the heuristics would just be an implementation detail). Even then, I see no proof why all states that can reach the termination state would be visited (which I feel is what “completeness” implies).

Response: The algorithm terminates as soon as a first valid path is found (this is inherent to the nondeterministic search specification). We have further clarified this point in section 3.3. We have also added a short footnote for readers not familiar with nondeterministic Turing Machines.